# Real-Time CDC Consultation during the COVID-19 Pandemic—United States, March–July, 2020

**DOI:** 10.3390/ijerph18147251

**Published:** 2021-07-06

**Authors:** Daniel Wozniczka, Hanna B. Demeke, Angela M. Thompson-Paul, Ugonna Ijeoma, Tonya R. Williams, Allan W. Taylor, Kathrine R. Tan, Michelle S. Chevalier, Elfriede Agyemang, Deborah Dowell, Titilope Oduyebo, Miriam Shiferaw, Sallyann M. Coleman King, Anna A. Minta, Katherine Shealy, Sara E. Oliver, Catherine McLean, Maleeka Glover, John Iskander

**Affiliations:** 1Clinician On-Call Team, COVID-19 Response, Centers for Disease Control and Prevention, Atlanta, GA 30333, USA; mln7@cdc.gov (H.B.D.); eup4@cdc.gov (A.M.T.-P.); wbv4@cdc.gov (U.I.); tis5@cdc.gov (T.R.W.); avt0@cdc.gov (A.W.T.); kit4@cdc.gov (K.R.T.); xde1@cdc.gov (M.S.C.); lwz6@cdc.gov (E.A.); gdo7@cdc.gov (D.D.); ydk7@cdc.gov (T.O.); wji3@cdc.gov (M.S.); fjq9@cdc.gov (S.M.C.K.); yzv1@cdc.gov (A.A.M.); srk3@cdc.gov (K.S.); yxo4@cdc.gov (S.E.O.); cvm9@cdc.gov (C.M.); mhg6@cdc.gov (M.G.); jxi0@cdc.gov (J.I.); 2Epidemic Intelligence Service, Centers for Disease Control and Prevention, Atlanta, GA 30333, USA; 3U.S. Public Health Service Commissioned Corps, Rockville, MD 20852, USA

**Keywords:** COVID-19, CDC, real-time consultation, clinician inquiries, health department inquiries

## Abstract

Context: In response to the COVID-19 pandemic, the Centers for Disease Prevention and Control (CDC) clinicians provided real-time telephone consultation to healthcare providers, public health practitioners, and health department personnel. Objective: To describe the demographic and public health characteristics of inquiries, trends, and correlation of inquiries with national COVID-19 case reports. We summarize the results of real-time CDC clinician consultation service provided during 11 March to 31 July 2020 to understand the impact and utility of this service by CDC for the COVID-19 pandemic emergency response and for future outbreak responses. Design: Clinicians documented inquiries received including information about the call source, population for which guidance was sought, and a detailed description of the inquiry and resolution. Descriptive analyses were conducted, with a focus on characteristics of callers as well as public health and clinical content of inquiries. Setting: Real-time telephone consultations with CDC Clinicians in Atlanta, GA. Participants: Health care providers and public health professionals who called CDC with COVID-19 related inquiries from throughout the United States. Main Outcome Measures: Characteristics of inquiries including topic of inquiry, inquiry population, resolution, and demographic information. Results: A total of 3154 COVID-19 related telephone inquiries were answered in real-time. More than half (62.0%) of inquiries came from frontline healthcare providers and clinical sites, followed by 14.1% from state and local health departments. The majority of inquiries focused on issues involving healthcare workers (27.7%) and interpretation or application of CDC’s COVID-19 guidance (44%). Conclusion: The COVID-19 pandemic resulted in a substantial number of inquiries to CDC, with the large majority originating from the frontline clinical and public health workforce. Analysis of inquiries suggests that the ongoing focus on refining COVID-19 guidance documents is warranted, which facilitates bidirectional feedback between the public, medical professionals, and public health authorities.

## 1. Introduction

The novel coronavirus disease 2019 (COVID-19) pandemic caused by severe acute respiratory syndrome coronavirus 2 (SARS-CoV-2) began in Wuhan, Hubei Province, China in December 2019, and spread to other countries, including the United States. In response, the U.S. Centers for Disease Control and Prevention (CDC) activated its Emergency Operations Center (EOC) on 20 January 2020. The EOC is responsible for coordinating emergency responses to domestic and international public health threats [1]. CDC deployed clinicians, epidemiologists, and other experts to the EOC or the field, to assist with case identification, contact tracing, evaluation of persons under investigation (PUI) for COVID-19, and medical management of COVID-19 cases [2,3].

Healthcare providers, public health practitioners, and organizations across the nation had many concerns as SARS-CoV-2 rapidly spread globally and in the United States. The lack of data and the uncertainty around the COVID-19 outbreak led to an urgent demand for guidance while efforts to improve our understanding of this novel virus continued. CDC responded by issuing several guidance documents and tools. CDC also deployed a multidisciplinary team of CDC clinicians within the EOC to provide real-time consultation to healthcare providers, public health practitioners, and health departments on the prevention and control of COVID-19. The team had expertise in infectious disease, internal medicine, obstetrics, pediatrics, health communications, pharmacology, and statistics.

Prior to March, clinicians primarily responded to inquiries related to travel-related COVID-19 cases, identifying persons under investigation (PUI), testing for SARSCoV-2, and advising the quarantine of repatriated travelers. As states began to identify community transmission of COVID-19, more guidance and tools were developed and the nature of inquiries from frontline health workers and health departments rapidly expanded. Consequently, CDC expanded the consultation in March 2020 to answer context-specific inquiries, assist in interpreting evolving CDC guidance, and coordinate technical assistance in real-time. We aim to describe the demographic and public health characteristics of inquiries related to the COVID-19 pandemic that CDC received by telephone from March to July 2020 and to highlight the successes, challenges, and utility of providing a 24/7 real-time consultation during a major public health event such as the COVID-19 pandemic. Lastly, we discuss lessons learned that can be applied in future epidemic response efforts.

## 2. Methods

### 2.1. Real-Time Consultation Structure

The EOC staff routed telephone inquiries related to COVID-19 to the multidisciplinary team of CDC clinicians, who provided real-time consultations in four overlapping shifts. The structure consisted of a triage team that assigned callers to clinicians, an escalation team to whom clinicians referred inquiries that required further technical assistance, a data team that monitored data collection, clinicians who were responsible for the real-time consultation, and medical directors that oversaw the overall operations. The clinician team categorized inquiries into two categories: (1) inquiries that could be answered using published guidance and tools on the CDC COVID-19 website; (2) inquiries that could not be answered either by publicly available documents or internal documents and required referral to a Subject Matter Expert (SME) in various task forces within CDC.

### 2.2. Data Collection

The inquiry database was created using a pre-existing secure, web-based data collection platform known as Data Collation and Integration for Public Health Event Response (DCIPHER). The database captured all COVID-19 related inquiries answered by the clinician team. Information captured included date of inquiry, inquirer’s contact information, inquirer’s affiliation, state, general topic of inquiry, individual patient information if the call pertained to a specific individual, or inquiry population if the call sought guidance for a more general group, a detailed description of the inquiry and resolution.

We defined inquirer affiliation as the type of facility or organization from which the call originated (e.g., clinical site, health department, Department of Defense). Inquiry population refers to the group for which guidance was sought (e.g., healthcare worker, pediatric, long term care facility, persons >65 years of age, pregnant). Inquiry topic refers to the subject for which caller was seeking guidance (e.g., preventing transmission, worker safety, testing, PUI determination, clinical management). This activity was reviewed by CDC and was conducted consistent with applicable federal law and CDC policy [4].

### 2.3. Data Analysis

We analyzed data collected from 11 March through 18 July 2020. Using the U.S. Department of Health and Human Services (HHS) defined regions, we examined geographic variation in call volume [5]. We compared weekly call volume with weekly incident COVID-19 cases in the U.S. reported to CDC over the same time period. Publicly available COVID-19 case counts were obtained from COVIDView, a weekly surveillance summary of COVID-19 activity posted on the CDC website [6]. We performed all data management and analyses using SAS^®^ 9.4 (SAS Institute Inc., Cary, NC, USA).

## 3. Results

CDC clinicians provided real-time remote consultation for a total of 3154 inquiries from healthcare providers, public health practitioners, and health departments on a range of topics. More than half of inquiries originated from clinical sites (62.0%), followed by health departments (14.1%) and the remaining inquiries (24.3%) came from federal agencies, long-term care facilities, businesses, and research organizations. Nearly one in four calls (24.2%) were concerns and questions about an individual patient. Health care workers (HCW) were the most common population about whom callers inquired (27.7%), followed by long term care facility residents (4.4%), non-HCW critical populations (3.5%), pediatric population (3.3%), persons ≥65 years of age (2.3%), and pregnant persons (1.0%) (Table 1).

Inquiries were complex and often involved several topic areas. Nearly half of calls (44.6%) concerned preventing SARS-CoV-2 transmission (including questions about infection control and personal protective equipment (PPE)). One quarter of calls concerned testing (25.6%) or worker safety guidance (24.9%). Infection prevention and control (19.9%), risk assessment (13.7%), isolation and quarantine guidance (13.2%), clinical management of COVID patients (12.8%), and return to work (12.7%) were among other topics frequently discusses during calls. Clinicians responded and resolved 78.0% of all inquiries during the initial phone call. The remaining 22.0% of inquiries were referred (urgently or non-urgently) to specific SMEs or another task force for coordination of response activity (Table 1).

Figure 1 shows weekly number of inquiries received and new COVID-19 cases in the U.S reported to CDC. The volume of inquiries fluctuated throughout the course of the pandemic and corresponded to changes in both CDC guidance and response efforts (Figure 1). The highest number of calls occurred in mid-March, with 236 inquiries in the week of 8–14 March, and 247 inquiries in 15–21 March. The number of weekly reported COVID-19 cases increased in the U.S from 1959 in the week of 8–14 March to 22,349 the week of 15–21 March. Incident COVID-19 cases remained between 150,000–200,000 cases per week from the week beginning 12 April through the week ending 20 June, when incident cases sharply increased each week until the last week of this study period. The volume of inquiries also varied by geographic location (Figure 2). The HHS region with the highest call volume (701 consultations) was Region 4, corresponding to the Southeast United States. The regions with the fewest consultations are Regions 7 and 8, with 87 and 97 consultations respectively, corresponding to the Midwest and Western areas.

## 4. Discussion

Providing real-time consultation and tracking inquiries and resolutions during a public health emergency serves multiple purposes. Real-time consultation provided direct on-demand support, tools, and timely technical assistance to healthcare workers and health departments [7]. The highest number of inquiries was received during the first two weeks following the United States declaring a State of Emergency for COVID-19 on March 13 [8]. At that time, information about SARS-CoV-2 diagnosis, testing, management, and prevention of transmission was limited. Guidance was developed and updated continuously as new evidence emerged. The release of multiple guidance and tools along with the community spread of SARS-CoV-2 and closing of most cities made it challenging for clinicians and health departments to interpret and implement rapidly evolving guidance appropriately. Challenges included the application of guidance to various patient populations or settings, using previous versions of guidance that had been recently updated, and dealing with clinical or public health practice scenarios for which guidance is not yet available.

Most inquiries came from clinical sites and from concerned HCWs. This finding indicates not only the large demand for real-time consultations with CDC clinicians during outbreaks such as COVID-19, but also further illustrates concerns of frontline healthcare workers about risk of exposure. Between February and June, 100,572 HCWs were reported to CDC as having contracted COVID-19 [9]. Clinicians inquiring about exposure risk were frequently calling about their own safety in addition to transmission risk to patients, as frequent inquiries included risk assessment after a known or potential exposure, implementing the correct return to work strategy for exposed personnel, isolation and quarantine guidance, and PPE use in healthcare setting guidance.

The number of inquiries was not clearly associated with the trend in the number of new COVID-19 cases in the U.S. However, the overall number of inquiries was consistent with the distribution of COVID-19 cases across HHS regions as Region 4 had the highest COVID-19 case counts, while Regions 7 and 8 had fewer cases [6]. Higher volume of inquiries in March 2020 were expected when most of the preparations and changes in healthcare and public health services were implemented in response to rising COVID-19 cases (Figure 1). For example, clinicians inquired about ensuring transmission prevention when their healthcare facility had their first confirmed COVID-19 case, or the first exposure risk to an HCW. This large volume of inquiries in March is consistent with increased number of calls that poison control centers and state level centers received regarding cleaners and disinfectants use related to COVID-19 [10,11] Over time, the volume steadily declined and plateaued for most part of the study period. However, there were times when a sharp increase in the number of inquiries occurred due to factors other than the increase in COVID-19 cases. For example, the rise in call volume in mid-May corresponded to (1) CDC’s website advertisement of the real-time remote consultations with CDC clinicians, and (2) CDC’s release of an official health advisory on Multisystem Inflammatory Syndrome in Children (MIS-C) [12], which resulted in inquiries seeking information on MIS-C diagnostic criteria and clinical management. Another similar spike in number of inquiries occurred during the week of 15 June when the CDC updated its guidance on infection prevention and control in dental settings (Figure 1).

Tracking inquiries and resolutions during a public health emergency is useful to facilitate bidirectional feedback on guidelines between the inquirers and CDC. This allows for more thorough responses to future inquiries, as well as identifying guidance that needed further clarification. For instance, near the start of the pandemic in February 2020, CDC published guidelines for risk stratification for potentially exposed healthcare workers. These guidelines were significant, as HCW’s risk of exposure determines when it may be safe for an HCW to continue seeing sick patients, or if they must self-isolate. As more evidence emerged and more inquiries were received, CDC continued to update the guidance with further details, including a 7 March 2020 update with allowances for asymptomatic healthcare workers who have had an exposure to continue to work after other options to improve staffing have been exhausted and a 15 April 2020 update to change the period of exposure risk from onset of symptoms to 48 h before symptom onset. The guidance was substantially streamlined and simplified on 19 May 2020. This updated new format to the CDC guidance was followed by an increase in inquiries regarding risk to healthcare workers, particularly how risk depends on whether or not the patient with COVID-19 was wearing a cloth face covering or facemask. The feedback from clinicians resulted in the risk exposure guidance being re-formatted on 23 May 2020 to ensure clarity. Another example was the need for clarification of varying temperature thresholds for fever in HCWs, for elderly individuals, and for the general population. As more inquiries came in, CDC clinicians recognized the need to include the reasoning behind these differences, which led to a harmonization of guidelines across populations for temperature thresholds. Another example of bidirectional feedback is the influx of inquiries received from dental offices in mid-June (Figure 1). Upon the release of updated guidance for dental settings on 15 June, there was widespread concern about the need for waiting periods between consecutive patients. Additional infection prevention guidance was developed for both non-COVID and COVID patients requiring dental care, to ensure clarity and alleviate the concerns of dental offices across the nation.

Another use of tracking and monitoring inquiries is the identification of emerging trends during outbreaks that may require development of new guidance, as well as collaboration and coordination between CDC task forces and local health departments. For example, recovered COVID-19 patients were asked to isolate for prolonged periods, sometimes months at a time. The CDC guidance in April through June recommended a test-based strategy, with at least two consecutive negative tests from respiratory specimens collected ≥24 h apart to discontinue isolation. Monitoring of inquiries highlighted how the test-based strategy could result in prolonged isolation especially for asymptomatic patients, which led to further investigations to determine if the test-based strategy should be recommended to all COVID-19 patients. A new guideline was released in July 2020, that no longer recommended the test-based strategy as the primary means to end isolation, as some patients continued to test positive for COVID-19 post-recovery even when they were likely no longer able to transmit SARS-CoV-2. Tracking and monitoring inquiries also facilitated the identification of areas or facilities requiring immediate assistance. The assistance can be either in the form of extensive collaboration with local health departments, or deployment of CDC personnel to the affected areas. These scenarios occurred during the spring of 2020, as during the real-time consultation with the CDC clinician, it would become evident that more than a single consultation was required. They resulted in CDC personnel being deployed to skilled nursing facilities, and correctional facilities around the country. Another example is the deployments to Native American reservations and collaboration with local and tribal leaders. Broad clinical and technical assistance was provided as a result of initial inquiries with clinicians on call. As of 25 July 2020, CDC deployed more than 200 teams to 55 state, tribal, local, and territorial health departments in response to requests for assistance with COVID-19 response [13].

Our findings are subject to at least three limitations. First, our data are from clinicians, public health workers, and health departments, and therefore not representative of inquiries from the general public. Inquiries from the general public are answered by CDC-INFO [14]. Secondly, this paper included only real-time consultation with CDC clinicians occurring as a result of phone call inquiries and did not include email inquiries that led to a later consultation. Thus, it does not represent the full scope of inquiries submitted to CDC. The nature and acuity of phone consultations could be different than email inquiries, as those requiring immediate assistance may have been more likely to arrive via phone than email. Third, some state and local health departments started their own hotline systems and responded to COVID-19 inquiries. Thus, this paper only describes inquiries that directly came to CDC at the national level and does not necessarily reflect those received at the state or local levels. The CDC may have been more likely to receive inquiries from states that did not offer real-time consultation at their respective health departments.

## 5. Conclusions

As part of CDC’s COVID-19 response, real-time consultation by CDC clinicians served an important public health surveillance function. It not only served to identify questions and concerns from key clinical and public health partners, but also provided feedback that led to modifications in CDC guidance and sparked creation of new resources. Large public health agencies may wish to consider tracking and analyzing inquiries as a way to maintain engagement with stakeholders, enhance quality improvement of materials they provide, and focus on clarifying guidance documents. Having staff members serve in this liaison role and work with partners on the frontlines also supports values of open communication and service to the public.

## Figures and Tables

**Figure 1 ijerph-18-07251-f001:**
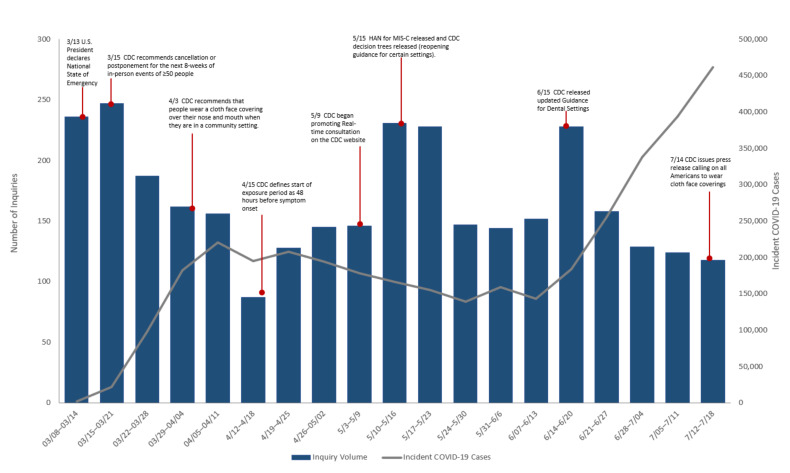
Weekly inquiries and COVID-19 incident cases—United States, 11 March–18 July 2020.

**Figure 2 ijerph-18-07251-f002:**
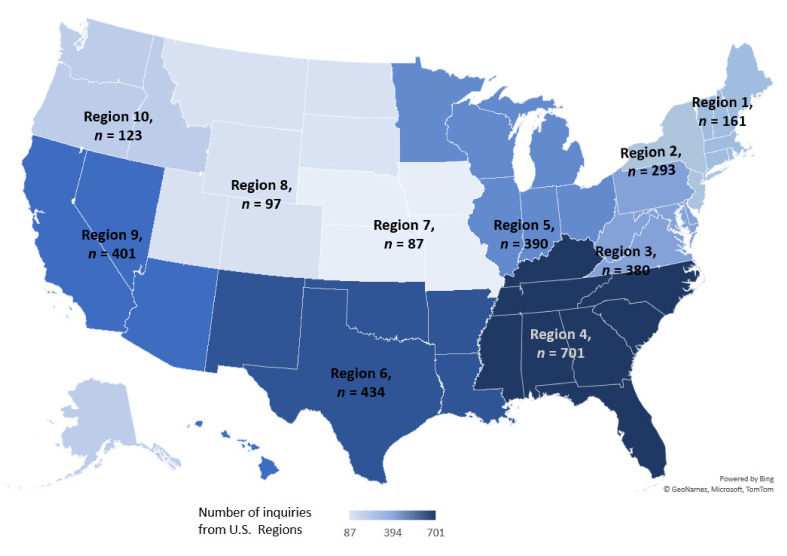
Total inquiries by Department of Health and Human Services Region—United States, 11 March–18 July 2020.

**Table 1 ijerph-18-07251-t001:** Summary of COVID-19 related telephone inquiries for real-time consultation—United States, 11 March–18 July 2020. (N = 3154).

Characteristics	N (%)
**Total**	3154 (100)
**Inquirer’s Affiliation (N = 3154)**	
Clinical Site	1956 (62.0)
Health Department	444 (14.1)
Other ^a^	765 (24.3)
**Inquiry Populations (N = 3154)**	
Individual patient ^b^	762 (24.2)
Health Care Workers (HCW)	873 (27.7)
Non-Health Care Critical Worker	111 (3.5)
Long Term Care Facility	140 (4.4)
Pediatric	104 (3.3)
Persons ≥65 years of age	71 (2.3)
Other ^c^	389 (12.3)
Unknown/Not documented	578 (18.3)
**Resolution (N = 3154)**	
Not escalated to another response task force	2461 (78.0)
Escalated to another response task force for resolution	693 (22.0)
Urgent	174 (5.5)
Non-Urgent	336 (10.7)
FYI	183 (5.4)
**Topics ^d^**	
Preventing Transmission	1406 (44.6)
COVID-19 Testing	808 (25.6)
Worker Safety	785 (24.9)
Infection Prevention and Control Guidance	629 (19.9)
Risk Assessment	433 (13.7)
Isolation or Quarantine Guidance	415 (13.2)
Clinical Management	405 (12.8)
Return To Work	402 (12.7)
Dental	327 (10.4)
Person Under Investigation Determination	298 (9.4)
Other ^e^	1515 (48.0)

^a^ Includes Department of Defense (*n* = 26), Federal Agency (*n* = 123), Long-Term Care Facilities (*n* = 87), Business (*n* = 86), Individual (*n* = 53), Non-Governmental Organization (41), Research/Academic Institution (*n* = 36), Not Documented/Unknown (*n* = 222). ^b^ Inquiries about individual patients referred to 309 symptomatic and 91 asymptomatic patients. Symptom status of other individual patients was unknown. ^c^ Includes Pregnant, Non-HCW Critical Workers, Tribal/Native American, Incarcerated, Behavioral Health Residents, Racial/ethnic minority, Homeless. ^d^ Up to three topics were identified for each call. ^e^ Other call topics included: Contact Tracing (*n* = 189), Postmortem (*n* = 158), Personal Protective Equipment (*n* = 238), Serology (*n* = 112), Reporting (*n* = 257), Disinfection Guidance (*n* = 75), Specimen Collection (*n* = 107), COVID Characteristics (*n* = 57), Vaccine & Therapeutics (*n* = 53), and all other (*n* = 269).

## Data Availability

Not Applicable.

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
