# Peer review of "Real-Time CDC Consultation during the COVID-19 Pandemic—United States, March–July, 2020"

_ijerph, 2021, doi:10.3390/ijerph18147251_

Round 1

Reviewer 1 Report

The paper reports on the initiative of the CDC to establish a real-time consultation telephone service to health care providers' inquiries on COVID-19. The methodology and the analysis of inquiries to answer the objective of the paper (i.e. demographics and characteristics of inquiries and correlation with COVID-19 epidemiology in US) are accurate. However, apart from allowing a bidirectional flow of information between health care providers and CDC, the impact of the findings on the response to the COVID-19 pandemic or to future outbreaks is limited. 

The major limitation of the article is to consider only the inquiries between March 11 and July 31, 2020, at a very early stage of the epidemic. It would have been of more interest to follow the evolving characteristics of inquiries with the introduction of vaccines or to evaluate the impact of CDC guidelines, health care measures adopted, and the consultation service offered by CDC clinicians.

In summary, this is a very accurate report of a meritorious and useful initiative, whose merit as a scientific article is disputable.

Author Response

Dear reviewer,

Thank you for your feedback.  We utilized all the data available to us at the time of writing this manuscript and getting it cleared. In addition, there were process changes in the way CDC conducted consultation that will make it difficult to merge data beyond the specified time frame. For this reason, we decided to stay with the time frame that corresponds to the first wave of the pandemic.

Reviewer 2 Report

Interesting work describing the realities of the current covid-19 pandemic. Results clear, well presented, discussion well conducted.

The paper I revieved was in my opinion new because of one reason - it describes the problem of e-health especially e-consultations in C-19 era. On one hand authors described rather technical layer than medical. Paper should be extended. The section about clinical purpose of consultations have to be added, especially anamnesis (diseases, clinical status, used medication, family anamnesis, previous physical activity, maybe sport activity), recommendations (what path of treatment, home, hospital, other, medicines). Additionally independent call – follow up at specified period should be performed to confirm adherence. Moreover every consultation have to be done by two independent physicians to calculate interobserver variability. If randomization and data blinding is possible there is possibility to calculate intraobserver variability too. It is possible but it is not necessary in basic, technical paper.

Author Response

Thank you for your comments. CDC clinicians were not the primary provider of patients but were there to provide support, tools, and technical assistance to the primary care providers. In addition, as we indicated in this report, only about a quarter of the inquiries were about individual patients. Other inquiries included questions about preventing transmission, risk assessment for exposure and return to work guidance. This is a cross-sectional analysis of data entered by clinicians who provided the real-time consultation independent of each other and  interobserver variability is not applicable in this paper.